# Retrospective Analysis of FED Method Treatment Results in 11–17-Year-Old Children with Idiopathic Scoliosis

**DOI:** 10.3390/children9101513

**Published:** 2022-10-03

**Authors:** Sandra Trzcińska, Kamil Koszela

**Affiliations:** 1Department of Physiotherapy, College of Rehabilitation in Warsaw, 01-234 Warsaw, Poland; 2Neuroorthopedics and Neurology Clinic and Polyclinic, National Institute of Geriatrics, Rheumatology and Rehabilitation, 02-637 Warsaw, Poland

**Keywords:** scoliosis, musculoskeletal disorders, spine, conservative treatment, rehabilitation

## Abstract

(1) Background: Idiopathic scoliosis is a major treatment problem due to its unknown origin and its three-dimensional nature. Attempts to cure it and search for new methods of physiotherapeutic treatment that would lead to its correction are one of the key issues of modern medicine. One of them is the fixation, elongation, de-rotation method (FED), used in the conservative treatment of idiopathic scoliosis. The aim of the study was evaluation of the short-term effectiveness of the FED method in the treatment of patients with idiopathic scoliosis. (2) Methods: Each patient underwent therapy based on the guidelines of the FED method. Patients were tested with the Bunnell scoliometer and the Zebris computer system. The treatment period was three weeks, after which the examinations were repeated. (3) Results: The results appeared to be statistically significant for all tested variables. (4) Conclusions: The examinations showed that the FED method had a statistically significant effect on the improvement of all parameters of posture examination, regardless of the size of the scoliotic deformation angle and bone maturity.

## 1. Introduction

Idiopathic scoliosis is a major treatment problem due to its unknown origin and its three-dimensional nature. Attempts to cure it and search for new methods of physiotherapeutic treatment that would lead to its correction are one of the key issues of modern medicine. 

One of the less known methods used in Poland for the treatment of patients with idiopathic scoliosis is the FED method (fixation, elongation, de-rotation) established in Spain. The FED treatment method uses a device where corrective forces act on the curvature. The strength of the device is focused on stabilizing, stretching, and de-rotating the spine, under the control of an innovative computer program [1]. As a method that uses a special apparatus for the correction of scoliotic deformity, it has been of great interest to medics in recent years. However, reports that assess the effects of its application are still insufficient [2].

There are several ways to test the effectiveness of the FED method in treating patients with idiopathic scoliosis. The most popular, but harmful due to radiation exposure, is a radiological examination, which is still the standard for diagnosing scoliosis [3,4,5]. Scoliosis patients can get 10 to 25 spinal X-rays over several years equating to a maximum 10 to 25 mGy of cumulative exposure. Patients who were diagnosed at a younger age and received early and ongoing treatment may be subjected to up to 40 to 50 X-rays, 50 mGy in total [6]. Other non-invasive devices have been used more and more frequently to assess the effects of therapeutic activities undertaken, including the assessment and comparison of various treatments and methods. Such a test can be both a clinical examination, including a scoliometer test as a simple and reliable device for measuring the transverse plane of the spine-trunk rotation, and a more modern three-dimensional posture test, such as the Zebris system. The Zebris system is a modern and specialized apparatus that uses ultrasound, enabling non-invasive examination of the body posture by creating a three-dimensional image of the patient’s figure. The system performs a computer analysis to create a report [7]. While the scoliometer is traditionally used to measure trunk rotation at the apex of the curvature, it can also be used to assess global trunk rotation, which evaluates the overall impact of therapy on the spine. A similar test assessing the overall effect of the therapy, but in the frontal plane, is the examination of scoliotic deformity with the Zebris system.

The aim of the study was to evaluate the results of the FED method in patients with idiopathic scoliosis in the short term.

## 2. Materials and Methods

### 2.1. Study Population 

The study included 81 subjects, 72 girls and 9 boys, aged from 11 to 17 (mean 14.28 ± 1.63). Each of the subjects had a current radiograph, which assessed the following: Risser test, the size of the Cobb angle for individual curvatures and the type of scoliosis was marked according to the King-Moe classification. All patients had idiopathic double-curve scoliosis of type I or II, characterized by the presence of a sigmoid curve in the thoracic (type I) or in the lumbar (type II) segments greater than the other one.

#### Inclusion Criteria

-Current X-ray scan (not older than 1 month) covering the pelvic girdle, diagnosed double-curve idiopathic scoliosis of type I and II according to King-Moe classification, with the Cobb angle between 10 and 60 degrees of primary scoliosis;-Age 11–17 years;-Incomplete ossification;-No contraindications to the therapy from other systems;-Consent to examination procedures.-Exclusion criteria:-Scoliosis of other than idiopathic origin;-Risser sign = 5–finished ossification;-Coexisting diseases of other organs that prevent participation in the program;-Lack of consent of the patient and the guardian to examinations and participation in the program.

### 2.2. Study Protocol

Each patient underwent therapy based on the guidelines of the FED method, which consisted of three basic elements: physical therapy as well as analytical and instrumental kinesiotherapy. The main component of the treatment was a special device that corrected the spine in 3 planes. With the help of a special vest, the patient was suspended in the device. Elongation was performed by a computer-controlled hoist, which, at the same time, regulated the pressure of the mobile arm correcting the apex of the scoliotic curve. Other arms stabilized the scoliotic curve at its ends. The pressure force was determined depending on the patient’s ability, up to a maximum of 100 kg. The time of the procedure was 30 min, the time of corrective pressure performed by the pneumatic movable arm was 20 s, and the break was 10 s. The arm corrected the curve both in the frontal and rotational planes, owing to the possibility of its angular positioning. In order to prepare the patient for the therapy in the device, the tissues were made more flexible and blood supply improved in the places to be subjected to the therapy, so, in this study, electrostimulation of the muscles on the convex side of the curve and thermal treatment, in which warm compresses were placed in the deformation concavity, were used. Both procedures lasted 15 min. Then the patient performed exercises for about 20–30 min, individually selected in accordance with the guidelines of the FED method. The selection of exercises was based on the King-Moe classification, which divides scoliosis into 5 types in terms of the location of scoliotic curvatures. In addition, patients wore the Boston brace every day for approximately 21–22 h a day, except for FED therapy (up to 3 h) and personal care. The Boston brace was made on the basis of a plaster cast and self-report was used. 

Each participant was tested with the Bunnell scoliometer and the Zebris computer system on the day before therapy began. The treatment period was 3 weeks, after which the examinations were repeated on the day the therapy completed. In the study, the scoliometer assessed both the trunk rotation angle at the apex of both scoliotic curves for the thoracic (ATR Th) and lumbar (ATR L) spine, and the total spine rotation using the SDR summing parameter, which consisted in summing the values of the rotation on both curves as positive values regardless of their direction. The computer examination assessed overall scoliotic deformation (SD) in the frontal plane. This parameter was the sum of the angles of tangents from the seventh cervical to the fifth lumbar vertebrae (C7-L5).

This study was conducted according to the guidelines of the Declaration of Helsinki, and approved by the Bioethics Committee for Scientific Research at the College of Rehabilitation in Warsaw, number 100/2022.

### 2.3. Data Analysis

In order to answer the research questions, statistical analyses were performed with the IBM SPSS Statistics 27 (Armonk, NY, USA). This was used to analyze basic descriptive statistics, with Shapiro–Wilk test, and Student’s t-test for dependent samples, and Wilcoxon test and Spearman’s rho correlation analysis. The level of significance was considered to be α = 0.05.

## 3. Results

### 3.1. Data Analysis

Most patients (66.7%) had type II scoliosis according to the King-Moe classification. The mean Cobb angle at the thoracic level [°] was 35.91 ± 10.43, and at the lumbar level it was 33.54 ±10.94. Detailed results are presented in Table 1 and Table 2.

### 3.2. Basic Descriptive Statistics with the Shapiro-Wilk Test

In the first step of the analysis, the distributions of quantitative variables were checked. For this purpose, basic descriptive statistics were calculated together with the Shapiro-Wilk test examining the normality of the distribution. The results of the analysis are presented in Table 3.

The results of the Shapiro-Wilk test appeared to be statistically significant for the lumbar rotation angle-ATR L [°] and for the scoliotic deformation angle-SD [°] both before and after the therapy. This meant that distributions of these variables differed from the normal distribution. However, in the case of the lumbar rotation angle-ATR L [°], the value of the skewness did not exceed the absolute value of one, which indicated that the asymmetry was insignificant. Therefore, the analyses for the trunk and lumbar rotation angles-ATR Th [°] and AR L [°], and for the sum of two rotations-SDR [°], were performed based on parametric tests. Yet, for the scoliotic deformation angle-SD [°], the skewness exceeded the value of one. A detailed analysis showed that this value of skewness resulted from the presence of two outliers (+ 3SD). Thus, for these variables, the analyses were based on non-parametric tests.

### 3.3. Comparison of the Value of the Trunk Rotation Angle-ATR Th [°], the Lumbar Rotation Angle-ATR L [°], the Sum of Two Rotations SDR [°] and the Angle of Scoliotic Deformation-SD [°], before and after the Therapy

In the next step, it was checked whether the applied therapy influenced values of the trunk rotation angle-ATR Th [°], the lumbar rotation angle-ATR L [°], the sum of two rotations-SDR [°] and the angle of scoliotic deformation-SD [°]. For this purpose, the Student’s *t*-test was performed for dependent samples and, in the case of the scoliotic deformation angle, the non-parametric Wilcoxon test was performed. The analyses were performed for all patients, taking into account the division into gender and type of scoliosis, based on the King-Moe classification (Table 4).

The results appeared to be statistically significant for all tested variables. The values of individual parameters, the trunk and lumbar rotation angles, the sum of two rotations and the scoliotic deformation angle, were significantly lower after the therapy than before. Differences in measurements were observed in both girls and boys, and in subjects with scoliosis types I and II. Each of the differences between the measurements before and after the therapy was significant.

### 3.4. Correlations between Parameters Measured with X-ray, Scoliometer and Scolioscan

In the next step, it was verified whether the measurements made with the use of a scoliometer and Zebris computer system correlated with each other. For this purpose, Spearman’s rho correlation analyses were performed. The parameters were compared before and after therapy and for all observations in general. The results are presented in Table 5.

The analysis showed six statistically significant correlations. For the measurements performed before the therapy, a significant correlation was observed only in the case of the sum of two rotations-SDR [°]. In the case of measurements performed after the therapy, statistically significant correlations concerned the trunk rotation angle-ATR Th [°] and again the sum of two rotations-SDR [°]. On the other hand, the analysis performed for all measurements in total showed that the angle of scoliotic deformation-SD [°] was significantly related to each of the results obtained with the use of the scoliometer. Parameters that were measured with the scoliometer were positively related to the scoliotic deformation angle, which was measured with the Zebris computer system. Three of the statistically significant correlations were weak and three were moderately strong.

### 3.5. Comparison of the Value of the Trunk Rotation Angle-ATR Th [°], the Lumbar Rotation Angle-ATR L [°], the Sum of Two Rotations SDR [°] and the Angle of Scoliotic Deformation-SD [°], before and after the Therapy Depending on the Dimensions of the Larger Curve

In the next step, it was verified whether, as a result of the therapy, in the groups distinguished by the dimensions of the larger curvature, the individual measured parameters changed, i.e., the trunk rotation angle-ATR Th [°], the lumbar rotation angle-ATR L [°], the sum of two rotations-SDR [°] and the angle of scoliotic deformation-SD [°]. The analyses were carried out in groups distinguished according to the value of the larger Cobb angle (at the level of the thoracic or lumbar spine). It was assumed that I° results were up to 24°, II°, 25–45°, and III°, above 45°. Due to the very small number of individuals with I° (*n* = 4), the analyses were performed only for those with II° and III°. The Student’s *t*-test was used for dependent samples, and, in the case of the scoliotic deformation angle, the non-parametric Wilcoxon test was used (Table 6).

The results were statistically significant for all variables in every group. The values of trunk and lumbar rotation angles [°] (ATR Th and ATR L), sum of two rotations [°] and scoliotic deformation angle-SD [°] were lower after the therapy compared to those before the therapy. In both groups and for all variables, the observed differences were strong.

### 3.6. Comparison of the Value of the Trunk Rotation Angle-ATR Th [°], the Lumbar Rotation Angle-ATR L [°], the Sum of Two Rotations SDR [°] and the Angle of Scoliotic Deformation-SD [°], before and after the Therapy Depending on the Risser Sign Grade

The results were statistically significant for all variables in every group. The values of trunk and lumbar rotation angles [°] (ATR Th and ATR L), sum of two rotations [°] and scoliotic deformation angle-SD [°] were lower after the therapy compared to those before the therapy. In both groups and for all variables, the values were lower after the therapy compared to those before the therapy independently of the Risser Sign Grade, the observed differences were strong (Table 7).

## 4. Discussion

The FED method is a relatively little-known method of treatment. It originated in Spain, where most of the scientific reports on its effectiveness come from [8,9,10,11]. Conducted studies have concerned mainly single cases or studies in smaller groups of patients. At the time when the method became fairly common in Poland, projects concerning larger research groups began to appear [12,13,14]. Today, many reports assessing its effectiveness in the treatment of patients with idiopathic scoliosis come from Poland. The number of centers using this type of treatment is constantly growing and the method has a large number of supporters. The reports that appear, especially in recent years, prove its effectiveness in the treatment of patients with idiopathic scoliosis [15].

Studies on the effectiveness of the FED method in the treatment of patients with idiopathic scoliosis show a significant improvement in all measured parameters, both in the examination with the scoliometer and with the Zebris system. These changes occurred both in the transverse and frontal planes. It turned out to be important to correct not only individual curves, but also the entire area of the spine in both tested planes. These studies indicated a positive effect on the entire spine, in contrast to the existing therapeutic view on the correction of one curve at the expense of deteriorating the other [16,17].

The use of traditional scoliometer examination and other posture parameters enable the assessment of the effect of the undertaken treatment procedures, mainly focusing not only on the diagnosis, but also on the effectiveness of individual methods and ways of the therapy. However, there are more and more reports indicating that basic examination (e.g., with a scoliometer) is insufficient. The impact of the therapy not only on the rotation of individual curves, but also on the global rotation, should also be evaluated. For this reason, summing parameters were used in the assessment of body posture. These are even less well known, but more and more often they are willingly used to evaluate therapy and its effect on the transverse plane [18,19,20]. A similar study, assessing the overall effect of therapy on the entire spine, in terms of the frontal plane, involved the scoliotic deformation parameter tested with the Zebris system. The conducted research demonstrated a correlation between these two parameters, despite the fact that they relate to two different transverse and frontal planes.

The project also investigated the impact of FED therapy on scoliotic deformations in terms of its size. The modified SOSORT angular division was used, which divides scoliosis on the basis of size assessed by the Cobb method on the radiograph. The studies showed a significant improvement in all parameters, regardless of the size of the scoliosis.

The examinations showed an improvement in all the tested parameters at various stages of bone maturity (Risser Sign Grade 1–4). Improvement was observed in all patients after the use of the FED method, regardless of the Risser Sign Grade.

The harmfulness of radiological examinations has more and more frequently led to the use of modern computer diagnostics to assess the posture [21,22,23,24]. These computer diagnostics allow for a harmless assessment of treatment stages, in terms of posture, especially since they correlate with radiographic images [25]. However, a distinction should be made here regarding when to use these two different examinations: radiological examinations should be used to diagnose and assess the progression of scoliotic deformity, and computer-based monitoring and evaluation should be used to diagnose and assess the treatment course. Computer-based methods are probably an alternative to radiology, but not in all respects. These two examinations differ from each other but should complement each other. Carrying out diagnostic activities on various grounds results in better control of scoliotic deformation progressing to impaired quality of life or to surgery, which often causes complications, and which we would like to avoid in the treatment of patients with idiopathic scoliosis [26].

Due to the nature of the therapy, the patients’ follow-up was short-term. However, it would be worth performing repeated examinations at a longer time interval. The FED method is relatively new in Poland, so such studies should be carried out, especially related to the assessment of the impact of the method on the sagittal plane of the posture. It requires further analyses supplemented with long-term follow-up in a larger group of patients. A 3-month and 6-month follow-up examination is planned.

## 5. Conclusions

All assessed parameters of posture examination, both with the scoliometer and the Zebris system, showed a statistically significant improvement in patients treated with the FED method. Statistical improvement occurred both in boys and girls, as well as in all types of scoliosis. The analysis showed the occurrence of statistically significant correlations between the parameters of the posture examination with a scoliometer and the computer examination. The examinations showed that the FED method had a statistically significant effect on the improvement of all parameters of posture examination, regardless of the size of the scoliotic deformation angle and bone maturity. However, due to the short observation time, this method requires further research with a long follow-up period.

## Figures and Tables

**Table 1 children-09-01513-t001:** Gender and type of scoliosis.

		*n*	*%*
Gender (*n* = 81)	Girls	72	88.90%
	Boys	9	11.10%
King-Moe Classification (*n* = 81)	Type I	27	33.30%
	Type II	54	66.70%

*n*—number.

**Table 2 children-09-01513-t002:** Age and X-ray analysis of indicators.

	*n*	$\bar{x}$	*Min.*	*Max.*
Age [years]	81	14.28 ± 1.63	11.00	17.00
Risser sign [score]	81	2.85 ± 0.94	1.00	4.00
Cobb angle at the thoracic level [°]	81	35.91 ± 10.43	13.00	56.00
Cobb angle at the lumbar level [°]	81	33.54 ± 10.94	10.00	59.00
King-Moe Classification Type I-Cobb angle at the lumbar level	27	40.48 ± 6.70	24.00	59.00
King-Moe Classification Type II-Cobb angle at the thoracic level	54	39.01 ± 7.69	16.00	53.00

*n*—number, *x*—mean, *SD*—standard deviation, *Min*—the lowest value, *Max*—the highest value.

**Table 3 children-09-01513-t003:** Basic descriptive statistics of the studied variables together with the Shapiro-Wilk test for normality.

	$\bar{x}$	*Me*	*Sk.*	*Kurt.*	*Min.*	*Max.*	*W*	*p*
**Before Therapy (*n* = 81)**								
The trunk rotation angle-ATR Th [°]	11.30 ± 4.63	11.00	0.08	−0.47	1.00	22.00	0.99	0.535
The lumbar rotation angle-ATR L [°]	7.75 ± 4.90	7.00	0.50	−0.44	0.00	19.00	0.95	0.003 *
Sum of two rotations-SDR [°]	19.05 ± 5.46	18.00	0.48	−0.11	8.00	33.00	0.97	0.066
Scoliotic deformation angle-SD [°]	37.06 ± 14.21	35.00	1.10	2.51	8.30	90.60	0.94	<0.001 *
**After Therapy (*n* = 81)**								
The trunk rotation angle-ATR Th [°]	8.51 ± 4.19	8.00	0.13	−0.34	0.00	18.00	0.98	0.430
The lumbar rotation angle-ATR L [°]	5.05 ± 4.18	4.00	0.88	0.08	0.00	16.00	0.91	<0.001 *
Sum of two rotations-SDR [°]	13.56 ± 4.69	13.00	0.21	−0.57	4.00	23.00	0.98	0.114
Scoliotic deformation angle-SD [°]	24.25 ± 12.09	21.80	1.42	3.33	0.00	69.80	0.88	<0.001 *

*n*—number, *x*—mean, *SD*—standard deviation, *Me*—median, *Sk*—skewness, Kurt—kurtosis, *Min*—the lowest value, *Max*—the highest value, *W*—Shapiro–Wilk test, *p*—level of significance, *—statistical significance.

**Table 4 children-09-01513-t004:** Results of Student’s *t*-test for dependent samples and Wilcoxon’s test comparing individual parameters measured with a scoliometer and Zebris, before and after therapy.

	Before Therapy	After Therapy	*t/Z*	*p*	95% *CI*		*d Cohena/* *r*
	$\bar{x}$	$\bar{x}$			*LL*	*UL*	
**Total (*n* = 81)**							
The trunk rotation angle-ATR Th [°]	11.30 ± 4.63	8.51 ± 4.19	15.23	<0.001 *	2.43	3.15	1.69
The lumbar rotation angle-ATR L [°]	7.75 ± 4.90	5.05 ± 4.18	12.55	<0.001 *	2.27	3.13	1.39
Sum of two rotations [°]	19.05 ± 5.46	13.56 ± 4.69	19.17	<0.001 *	4.92	6.06	2.13
Scoliotic deformation angle-SD [°]	37.06 ± 14.21	24.25 ± 12.09	−7.40	<0.001 *	10.63	14.99	0.58
**Girls (*n* = 72)**							
The trunk rotation angle- ATR Th [°]	11.03 ± 4.63	8.26 ± 4.18	14.01	<0.001 *	2.37	3.16	1.65
The lumbar rotation angle-ATR L [°]	7.82 ± 4.94	5.07 ± 4.25	11.68	<0.001 *	2.28	3.22	1.38
Sum of two rotations [°]	18.85 ± 5.47	13.33 ± 4.73	17.70	<0.001 *	4.89	6.13	2.09
Scoliotic deformation angle-SD [°]	37.49 ± 13.83	24.38 ± 12.61	−6.96	<0.001 *	10.83	15.38	0.58
**Boys (*n* = 9)**							
The trunk rotation angle-ATR Th [°]	13.44 ± 4.28	10.44 ± 4.00	6.00	<0.001 *	1.85	4.15	2.00
The lumbar rotation angle-ATR L [°]	7.22 ± 4.84	4.89 ± 3.82	4.95	0.001 *	1.25	3.42	1.65
Sum of two rotations [°]	20.67 ± 5.39	15.33 ± 4.12	7.54	<0.001 *	3.70	6.96	2.51
Scoliotic deformation angle-SD [°]	33.62 ± 17.46	23.16 ± 6.89	−2.67	0.008 *	1.63	19.30	0.63
**Scoliosis type I (*n* = 27)**							
The trunk rotation angle-ATR Th [°]	9.00 ± 4.09	6.33 ± 3.45	7.45	<0.001 *	1.93	3.40	1.43
The lumbar rotation angle-ATR L [°]	9.67 ± 4.84	7.07 ± 4.20	9.66	<0.001 *	2.04	3.14	1.86
Sum of two rotations [°]	18.67 ± 6.00	13.41 ± 4.94	11.73	<0.001 *	4.34	6.18	2.26
Scoliotic deformation angle-SD [°]	35.57 ± 12.26	22.32 ± 9.24	−4.45	<0.001 *	9.29	17.20	0.61
**Scoliosis type II (*n* = 54)**							
The trunk rotation angle-ATR Th [°]	12.44 ± 4.49	9.59 ± 4.13	13.55	<0.001 *	2.43	3.27	1.84
The lumbar rotation angle-ATR L [°]	6.80 ± 4.69	4.04 ± 3.82	9.34	<0.001 *	2.17	3.35	1.27
Sum of two rotations [°]	19.24 ± 5.22	13.63 ± 4.60	15.22	<0.001 *	4.87	6.35	2.07
Scoliotic deformation angle-SD [°]	37.81 ± 15.14	25.21 ± 13.26	−5.95	<0.001 *	9.90	15.29	0.57

*n*—number, *x*—mean, *SD*—standard deviation, *t*—*t*-test, *Z*—Wilcoxon test, *p*—level of significance, 95% *CI*—95% Confidence Interval, *LL*—Lower Limit, *UL*—Upper Limit, *d Cohena*—effect size for Student’s *t* test for dependent samples, *r*—effect size for Wilcoxon test, *—statistical significance.

**Table 5 children-09-01513-t005:** Correlation analysis for measurements made with a scoliometer and Zebris computer system.

Measurements Made with a Scoliometer		Scoliotic Deformation Angle-SD [°]- Measurement Made with Zebris System		
		Before Therapy (*n* = 81)	After Therapy (*n* = 81)	Total (*n* = 162)
The trunk rotation angle-ATR Th [°]	*rho* Spearman	0.21	0.24	0.33
	significance	0.06	0.03 *	<0.001 *
The lumbar rotation angle-ATR L [°]	*rho* Spearman	0.16	0.13	0.23
	significance	0.16	0.26	0.003 *
Sum of two rotations [°]	*rho* Spearman	0.28	0.32	0.44
	significance	0.01 *	0.004 *	<0.001 *

*n*—number, *—statistical significance.

**Table 6 children-09-01513-t006:** Results of Student’s *t*-test for dependent samples and Wilcoxon’s test-comparison of individual parameters measured with the scoliometer and Zebris, before and after therapy, with division into groups distinguished on the basis of the value of greater Cobb angle.

	Before Therapy	After Therapy	*t/Z*	*p*	95% *CI*		*d Cohena/r*
	$\bar{x}$	$\bar{x}$			*LL*	*UL*	
**II°—Value of Greater Cobb Angle 25–45° (*n* = 56)**							
The trunk rotation angle-ATR Th [°]	10.34 ± 4.36	7.91 ± 3.95	12.99	<0.001 *	2.05	2.80	1.74
The lumbar rotation angle-ATR L [°]	7.18 ± 4.84	4.64 ± 4.05	10.05	<0.001 *	2.03	3.04	1.34
Sum of two rotations [°]	17.52 ± 3.90	12.55 ± 3.60	16.07	<0.001 *	4.35	5.58	2.15
Scoliotic deformation angle-SD [°]	32.92 ± 10.25	20.94 ± 8.91	−6.11	<0.001 *	9.56	14.41	0.58
**III°—Value of Greater Cobb Angle > 45° (*n* = 21)**							
The trunk rotation angle-ATR Th [°]	14.14 ± 4.41	10.24 ± 4.40	10.18	<0.001 *	3.10	4.70	2.22
The lumbar rotation angle-ATR L [°]	9.43 ± 5.08	6.43 ± 4.55	6.34	<0.001 *	2.01	3.99	1.38
Sum of two rotations [°]	23.57 ± 6.64	16.67 ± 5.79	11.48	<0.001 *	5.65	8.16	2.51
Scoliotic deformation angle-SD [°]	49.34 ± 16.89	32.85 ± 15.20	−3.88	<0.001 *	11.21	21.77	0.60

*n*—number, *x*—mean, SD—standard deviation, *t*—*t*-test, *Z*—Wilcoxon test, *p*—level of significance, 95% *CI*—95% Confidence Interval, *LL*—Lower Limit, *UL*—Upper Limit, *d Cohena*—effect size for Student’s *t* test for dependent samples, *r*—effect size for Wilcoxon test, *—statistical significance.

**Table 7 children-09-01513-t007:** Results of Student’s *t*-test for dependent samples and Wilcoxon’s test-comparison of individual parameters measured with the scoliometer and Zebris, before and after therapy, with division into groups distinguished on the basis of Risser Sign Grade.

	Before Therapy	After Therapy	*t/Z*	*p*	95% *CI*		*d Cohena* */r*
	$\bar{x}$	$\bar{x}$			*LL*	*UL*	
**Risser Sign Grade 1 (*n* = 9)**							
The trunk rotation angle-ATR Th [°]	11.89 ± 3.33	8.89 ± 4.14	4.24	0.002 *	1.37	4.63	1.41
The lumbar rotation angle-ATR L [°]	8.44 ± 5.41	6.56 ± 4.77	3.69	0.006 *	0.71	3.07	1.23
Sum of two rotations [°]	20.33 ± 7.50	15.44 ± 6.69	5.82	<0.001 *	2.95	6.83	1.94
Scoliotic deformation angle-SD [°]	48.14 ± 14.10	33.73 ± 13.38	−2.67	0.008 *	8.67	20.15	0.63
**Risser Sign Grade 2 (*n* = 15)**							
The trunk rotation angle-ATR Th [°]	12.40 ± 5.14	9.67 ± 4.78	7.12	<0.001 *	1.91	3.56	1.84
The lumbar rotation angle-ATR L [°]	7.40 ± 5.28	4.47 ± 4.24	5.12	<0.001 *	1.70	4.16	1.32
Sum of two rotations [°]	19.80 ± 5.20	14.13 ± 3.93	8.16	<0.001 *	4.18	7.16	2.11
Scoliotic deformation angle-SD [°]	39.16 ± 10.82	26.83 ± 11.69	−3.41	<0.001 *	8.13	16.53	0.62
**Risser Sign Grade 3 (*n* = 36)**							
The trunk rotation angle-ATR Th [°]	10.36 ± 4.82	7.83 ± 4.05	10.10	<0.001 *	2.02	3.04	1.68
The lumbar rotation angle-ATR L [°]	8.17 ± 5.28	5.31 ± 4.68	8.34	<0.001 *	2.16	3.56	1.39
Sum of two rotations [°]	18.53 ± 5.60	13.14 ± 4.77	12.38	<0.001 *	4.51	6.27	2.06
Scoliotic deformation angle-SD [°]	34.87 ± 15.15	21.10 ± 11.18	−5.23	<0.001 *	10.83	16.70	0.62
**Risser Sign Grade 4 (*n* = 21)**							
The trunk rotation angle-ATR Th [°]	11.86 ± 4.37	8.67 ± 4.10	8.10	<0.001 *	2.37	4.01	1.77
The lumbar rotation angle-ATR L [°]	7.00 ± 3.86	4.38 ± 2.84	7.11	<0.001 *	1.85	3.39	1.55
Sum of two rotations [°]	18.86 ± 4.61	13.05 ± 4.12	10.24	<0.001 *	4.63	6.99	2.23
Scoliotic deformation angle-SD [°]	34.57 ± 13.06	23.73 ± 11.61	−3.13	0.002 *	4.60	17.07	0.48

*n*—number, *x*—mean, *SD*—standard deviation, *t*—*t*-test, *Z*—Wilcoxon test, *p*—level of significance, 95% *CI*—95% Confidence Interval, *LL*—Lower Limit, *UL*—Upper Limit, *d Cohena*—effect size for Student’s *t* test for dependent samples, *r*—effect size for Wilcoxon test, *—statistical significance.

## Data Availability

The datasets analyzed during the current study are available from the corresponding author upon reasonable request.

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
