# Peer review of "Retrospective Analysis of FED Method Treatment Results in 11–17-Year-Old Children with Idiopathic Scoliosis"

_children, 2022, doi:10.3390/children9101513_

Round 1

Reviewer 1 Report

Thank you for your work and submitting. There are a lot of variables going on with this paper so hare to determine if one aspect has more impact that other aspects or if they are equal and its the totality of the program.  

Also, these are very short term results. Its similar to the initial in brace x-ray in that it shows the immediate impact.  I would like the authors to state that more clearly and state how predictive this type of early result is in the long term outcome for the patient.

In the abstract please define the letters FED (the name of the program) first, then use the acronym.   Also state these are short term results. Reference the SRS/SOSORT guidelines on publications - (Negrini, S.; Boards, S.; Hresko, T.M.; O’Brien, J.P.; Price, N.; SRS Non-Operative Committee. Recommendations for research studies on treatment of idiopathic scoliosis: Consensus 2014 between SOSORT and SRS non–operative management committee. Scoliosis 2015, 10, 1–12)

You state several times the harmful effects of radiation - please expand this and discuss the amount of exposure thought to be problematic.

Why is the author not using the SRS/SOSORT inclusion criteria.(see study reference above) 

You mention the Boston Brace - was this custom from scan, measurements only? Are you able to discuss the in brace correction?  Training of the orthotist.?  Please discuss compliance - was it objectively recorded with a thermal sensor or self report?

You describe multiple modalities.  What was the frequency? Daily? for how long each day. Were the patients provided with a home program? Move the fact that this was a camp type set up to the methods section.  

What is the follow up with the patients after the three week program?  

I think these results are very preliminary and hard to draw any conclusion on long term results. This needs to be stated.  

Author Response

Dear Reviewer,

Thank you very much for your review and valuable comments.

Thank you for your work and submitting. There are a lot of variables going on with this paper so hare to determine if one aspect has more impact that other aspects or if they are equal and its the totality of the program.

Thank you so much.

Also, these are very short term results. Its similar to the initial in brace x-ray in that it shows the immediate impact.  I would like the authors to state that more clearly and state how predictive this type of early result is in the long term outcome for the patient.

We agree with the reviewer. We've added this to the limitations section. Another 3 and 6-month follow-up study is planned.

In the abstract please define the letters FED (the name of the program) first, then use the acronym.   Also state these are short term results. Reference the SRS/SOSORT guidelines on publications - (Negrini, S.; Boards, S.; Hresko, T.M.; O’Brien, J.P.; Price, N.; SRS Non-Operative Committee. Recommendations for research studies on treatment of idiopathic scoliosis: Consensus 2014 between SOSORT and SRS non–operative management committee. Scoliosis 2015, 10, 1–12).

We've improved the abstract as suggested. 

Patients aged 11-17 years with idiopathic scoliosis with a cobba angle of 10-60 degrees were analyzed.

According to the guidelines in the article (Negrini, S.; Boards, S.; Hresko, T.M.; O’Brien, J.P.; Price, N.; SRS Non-Operative Committee. Recommendations for research studies on treatment of idiopathic scoliosis: Consensus 2014 between SOSORT and SRS non–operative management committee. Scoliosis 2015, 10, 1–12), they were divided into groups according to the Cobb angle: 10-24, 25-45 and more degrees. It was presented in tables. 

You state several times the harmful effects of radiation - please expand this and discuss the amount of exposure thought to be problematic.

We have added information about the harmful effects of radiation and references No 6.

Why is the author not using the SRS/SOSORT inclusion criteria.(see study reference above).

The inclusion criteria were more general, but we divided patients into Cobb angle and Risser Sign Grade groups according to the SRS/SOSORT criteria. It was presented in tables (No 6 and 7) in manuscript. 

You mention the Boston Brace - was this custom from scan, measurements only? Are you able to discuss the in brace correction?  Training of the orthotist.?  Please discuss compliance - was it objectively recorded with a thermal sensor or self report?

Boston Brace was not custom from scan, it was made on the basis of a plaster cast. Self report was used. We added this information to manuscript. 

You describe multiple modalities.  What was the frequency? Daily? for how long each day. Were the patients provided with a home program? Move the fact that this was a camp type set up to the methods section.  

Patients wore the Boston brace everyday for approximately 21-22 hours a day, except for FED therapy (up to 3 hours) and personal care. The patients were provided with a home program. We added this informations to M&M section.

What is the follow up with the patients after the three week program?  

The patient was re-examined after 3 weeks. Then the control was about a month later. Our study doesn't include a monthly inspection.

I think these results are very preliminary and hard to draw any conclusion on long term results. This needs to be stated.

We've added this to the limitations section. Another 3 and 6-month follow-up study is planned. We have corrected the conclusions.

Reviewer 2 Report

I commend the authors from this contribution in the time and effort spent in performing this analysis.  The authors describe the utilization of a relatively new technique for treatment of scoliosis, the FED fixation, elongation, derotation treatment.  They first recorded a wide variety of parameters utilized to measure the degree of scoliosis in multiple planes.  They then subject the patient sample to 3 weeks of treatment, and then they performed another measurement after the last treatment session. 

They find significant improvements in multiple scoliotic metrics through utilization of this treatment algorithm.

Overall this the study does provide some value and that they have included a vast number of different metrics of scoliosis which is far more complete than looking only at coronal plane deformity.  However, there are number of issues with the manuscript.  Primarily, the English language translation is very poor.  I have compiled a list of just a very small number of some of the obvious issues which will be clear to an English reader at the bottom of this text.

The instruction and methodologies are generally fine, although I would like to see the statistical analysis portion were flushed out.  The authors have decided to include much of the statistical reasoning in the results, for example using Wilcoxson rank sum for dependent samples, etc. but some of this should be included in the actual statistical analysis portion of things.  It is really incomplete as it stands.

The discussion also has a number of English language edits that need to be made and overall I think the contribution of this manuscript needs to be flushed out.  Discussion needs some serious work.

My biggest comment with regards to the actual quality of the study, however, has to do with the fact that the authors chose to perform the post therapeutic measurements immediately after the final lengthening.  Thus, it is unclear whether the improvements are transient or would actually still be the substantial several weeks after treatment.  I really think this limits the applicability of the study.  Obviously after a distraction exercise, much of the parameters would be improved, but does this actually hold up long term?

I would be happy to review after English language edits.

English language issues:

 "has shown great interest among medics in recent years."   "The time of the procedure was 30 minutes, the time of corrective pressure performed by the pneumatic movable arm - 20 seconds, and the break was 10 seconds. The arm corrected the curve both in the frontal and rotational plane, owing to the possibility of its angular positioning. In order to prepare the patient for therapy in the device, make the tissues more flexible and blood supplied in the places subjected to the therapy, electrostimulation of the muscles on the convex side of the curve was used and thermal treatment "   ."The analyzes were carried out for all patients and taking into account the division into."   "Due to the camp nature of the treatment of patients, the observations were carried out as short-term"   "The FED method is a relatively new method of treatment in Poland, so in the future, such studies would be recommended, especially in the assessment of the impact of the method also on the sagittal plane of the posture"      

Author Response

Dear Reviewer,

Thank you very much for your review and valuable comments.

I commend the authors from this contribution in the time and effort spent in performing this analysis.  The authors describe the utilization of a relatively new technique for treatment of scoliosis, the FED fixation, elongation, derotation treatment.  They first recorded a wide variety of parameters utilized to measure the degree of scoliosis in multiple planes.  They then subject the patient sample to 3 weeks of treatment, and then they performed another measurement after the last treatment session. 

They find significant improvements in multiple scoliotic metrics through utilization of this treatment algorithm.

Thank you so much.

Overall this the study does provide some value and that they have included a vast number of different metrics of scoliosis which is far more complete than looking only at coronal plane deformity.  However, there are number of issues with the manuscript.  Primarily, the English language translation is very poor.  I have compiled a list of just a very small number of some of the obvious issues which will be clear to an English reader at the bottom of this text.

Thank you so much. English correction has been made by english translator.

The instruction and methodologies are generally fine, although I would like to see the statistical analysis portion were flushed out.  The authors have decided to include much of the statistical reasoning in the results, for example using Wilcoxson rank sum for dependent samples, etc. but some of this should be included in the actual statistical analysis portion of things.  It is really incomplete as it stands.

The FED method has existed in Poland since 2010. Requires additional research. In our article we present an analysis of many parameters, the statistics are extensive but, in my opinion, necessary.

The discussion also has a number of English language edits that need to be made and overall I think the contribution of this manuscript needs to be flushed out.  Discussion needs some serious work.

English correction has been made by english translator.

My biggest comment with regards to the actual quality of the study, however, has to do with the fact that the authors chose to perform the post therapeutic measurements immediately after the final lengthening.  Thus, it is unclear whether the improvements are transient or would actually still be the substantial several weeks after treatment.  I really think this limits the applicability of the study.  Obviously after a distraction exercise, much of the parameters would be improved, but does this actually hold up long term?

We agree with the reviewer. We've added this to the limitations section. Another 3 and 6-month follow-up study is planned.

I would be happy to review after English language edits.

English language issues:

 "has shown great interest among medics in recent years."   "The time of the procedure was 30 minutes, the time of corrective pressure performed by the pneumatic movable arm - 20 seconds, and the break was 10 seconds. The arm corrected the curve both in the frontal and rotational plane, owing to the possibility of its angular positioning. In order to prepare the patient for therapy in the device, make the tissues more flexible and blood supplied in the places subjected to the therapy, electrostimulation of the muscles on the convex side of the curve was used and thermal treatment "   ."The analyzes were carried out for all patients and taking into account the division into."   "Due to the camp nature of the treatment of patients, the observations were carried out as short-term"   "The FED method is a relatively new method of treatment in Poland, so in the future, such studies would be recommended, especially in the assessment of the impact of the method also on the sagittal plane of the posture"     

English correction has been made by english translator.

Round 2

Reviewer 2 Report

I thank the authors for their re submission of this paper. Overall, they have improved upon it somewhat. I still have a number of concerns with the English language editing. Line numbers haven't been given on this revision, and I cannot comment specifically, but for example in the discussion, the following statements just really need improvement in clarity for the average English reader. We have to hold a high standard for publication in my opinion, especially for a journal with an impact factor of almost 3, and so I cannot right now recommend this article for publication. Extensive English language editing is still required throughout to make sure that the writing is up to par. I do commend the authors on their attempts to explain the limitations of the publication and I think as far as that goes from the study is now publishable.

“The use of both the traditional scoliometer examination and other parameters of posture assessment allows more and more frequently to assess the effect of the undertaken treatment steps”

“However, apart from standards, there are more and more reports extending the basic examination, for example with a scoliometer, to include an insight into the overall impact of therapy not only on the rotation of individual curves, but also on the overall rotation. Hence the use of summing parameters in the assessment of body posture.”

Author Response

Dear Reviewer,

Thank you so much for your comments.
The article has been checked by an English translator.
We have made changes as suggested.